# Nutrition and Wound Healing: An Overview Focusing on the Beneficial Effects of Curcumin

**DOI:** 10.3390/ijms20051119

**Published:** 2019-03-05

**Authors:** Martina Barchitta, Andrea Maugeri, Giuliana Favara, Roberta Magnano San Lio, Giuseppe Evola, Antonella Agodi, Guido Basile

**Affiliations:** 1Department of Medical and Surgical Sciences and Advanced Technologies “GF Ingrassia”, University of Catania, Via S. Sofia 87, 95123 Catania, Italy; martina.barchitta@unict.it (M.B.); andreamaugeri88@gmail.com (A.M.); giuliana.favara@gmail.com (G.F.); robimagnano@gmail.com (R.M.S.L.); 2General and Emergency Surgery Department, Garibaldi Hospital, Piazza Santa Maria di Gesù, 95100 Catania, Italy; giuseppe_evola@hotmail.it; 3Department of General Surgery and Medical-Surgical Specialties, University of Catania, Via Plebiscito 628, 95124 Catania, Italy; gbasile@unict.it

**Keywords:** wound, wound healing, diet, nutrition, micronutrients, macronutrients, curcumin, amino-acids, vitamins, minerals

## Abstract

Wound healing implicates several biological and molecular events, such as coagulation, inflammation, migration-proliferation, and remodeling. Here, we provide an overview of the effects of malnutrition and specific nutrients on this process, focusing on the beneficial effects of curcumin. We have summarized that protein loss may negatively affect the whole immune process, while adequate intake of carbohydrates is necessary for fibroblast migration during the proliferative phase. Beyond micronutrients, arginine and glutamine, vitamin A, B, C, and D, zinc, and iron are essential for inflammatory process and synthesis of collagen. Notably, anti-inflammatory and antioxidant properties of curcumin might reduce the expression of *tumor necrosis factor alpha* (*TNF-α*) and *interleukin-1* (*IL-1*) and restore the imbalance between reactive oxygen species (ROS) production and antioxidant activity. Since curcumin induces apoptosis of inflammatory cells during the early phase of wound healing, it could also accelerate the healing process by shortening the inflammatory phase. Moreover, curcumin might facilitate collagen synthesis, fibroblasts migration, and differentiation. Although curcumin could be considered as a wound healing agent, especially if topically administered, further research in wound patients is recommended to achieve appropriate nutritional approaches for wound management.

## 1. Introduction

Wound healing implicates a well-orchestrated complex of biological and molecular events that involve cell migration, cell proliferation, and extracellular matrix deposition. Although these processes are similar to those driving embryogenesis, tissue and organ regeneration, and even pathological conditions [1,2], certain differences exist between adult wounds and these other systems. In acute wounds—cutaneous injuries that do not have an underpinning pathophysiological defect—the main evolutionary force may have been to achieve repair quickly and with the smallest amount of energy [2]. In contrast, evolutionary adaptations have probably not occurred in chronic wounds with pre-existing pathophysiological abnormalities, resulting in impaired healing [3]. Wound care places an enormous drain on healthcare resources worldwide. For instance, in the United States, it has been estimated that 3% individuals over 65 years will have a wound at any one time [4], with an estimated cost to the healthcare system of approximately US $25 billion each year [5]. In low-income countries, an even higher incidence, due to traumatic injuries and ulcers, is expected. Recently, the World Health Organization (WHO) has recognized the unmet need for an interdisciplinary approach facing this global challenge, which has been accordingly addressed by the Association for the Advancement of Wound Care (AAWC) Global Volunteers program [6].

Despite strides in technological innovations of a wide range of treatments against wounds, non-healing wounds continue to challenge physicians. Hence, further efforts are needed to improve our scientific understanding of the repair process and how that knowledge can be used to develop new approaches to treatment. Malnutrition is a common risk factor that might contribute to impaired wound healing [7,8,9]. In recent years, several lines of evidence have pointed out biochemical and molecular effects of several nutrients in the wound healing process, supporting the notion that a complementary nutritional approach might be useful in wound treatment, especially for chronic non-healing wounds [10]. Here, we provide an overview of biological and molecular events in wound healing and the effects of malnutrition and specific nutrients on this process (search strategy and selection criteria are shown in Figure 1). In line with the current Special Issue “Curcumin in Health and Disease”, we have also focused on beneficial effects and related molecular mechanisms of curcumin—a natural phenol from the rhizome of *Curcuma longa*—which might enhance healing processes via antioxidant and anti-inflammatory properties [11]. In fact, curcumin was commonly used in traditional medicine for the treatment of biliary and hepatic disorders, cough, diabetic ulcers, rheumatism and sinusitis [11]. More recently, curcumin has been investigated extensively as an anti-cancer [12], anti-aging [13], and wound healing agent [11]. For instance, it has been demonstrated the beneficial effect of curcumin on the progression of endometriosis, a common disorder affecting women during reproductive age which shares some molecular events with wound healing (i.e., adhesion and proliferation, cellular invasion and angiogenesis) [14]. To date, most of the current knowledge on wound healing derives from in vitro and in vivo studies, while epidemiological investigations are scarce. To solve the question of whether curcumin is a suitable wound healing agent, we have summarized its biochemical and molecular effects during the different phases of wound healing, as well as evidence from epidemiological studies. 

### 1.1. The Wound Healing Process and Impaired Healing

The type, size, and depth of wounds have significant repercussions on cellular and molecular events that occur after cutaneous injury. As reviewed by Falanga [15], it is useful to divide the wound healing process into four overlapping steps of coagulation, inflammation, migration-proliferation, and remodeling (Figure 2). While acute wounds show a linear progression of these overlapping events, the progression in chronic wounds does not occur in synchrony, with some areas being in different phases at the same time [15].

In the first phase after injury, the formation of a fibrin plug (i.e., an aggregate of platelets, fibrinogen, fibronectin, vitronectin, and thrombospondin) is necessary both for hemostasis and for covering and protecting the wound from bacteria [2,16]. Beyond that, fibrin plug also provides an extracellular matrix for cell migration [2] and releases growth factors (e.g., platelet-derived growth factor—PDGF—and transforming growth factor—TGF) involved in the recruitment of cells to the wound [1,2]. In the inflammatory phase, endothelial expression of selectins slows down leukocytes in the bloodstream, so as to enable them to move through endothelial gaps by binding to integrins into the extracellular area [1]. Neutrophils and macrophages recruited to the wound remove foreign particles and produce a wide range of growth factors and cytokines that promote fibroblast migration and proliferation [17]. Hypoxia—which occurs immediately after injury—is one of the main triggers of keratinocyte migration, angiogenesis, fibroblast proliferation, and the releasing of growth factors and cytokines (i.e., PDGF, vascular endothelial growth factor, and TGF) [18]. Later, fibroblasts and endothelial cells form the early granulation tissue that begins the processes of wound contraction, which in turn is an efficient driver of wound closure [2]. Extracellular matrix proteins are crucial in this phase because they provide substrates for cell migration and structures that restore the function and integrity of the tissue [18]. The formation of new blood vessels re-establishes tissue perfusion, allowing for the re-supply of oxygen and other nutrients [17]. Finally, once closure of wound has been achieved, remodeling of the resulting scar takes places over weeks or months, with a reduction of both cell content and blood flow, degradation of extracellular matrix, and further contraction and tensile strength [15]. 

While venous or arterial insufficiency, diabetes, and local-pressure are the most common pathophysiological causes of wounds and ulcers, several local and systemic factors can impair wound healing. The first ones consist of the presence of foreign bodies, tissue maceration, ischemia, and infection. The second ones include aging, malnutrition, diabetes, and renal diseases. In addition to these factors, a reduction in active growth factors may partially explain why certain wounds fail to heal. Chronic ulcers seem to have reduced levels of PDGF, TGF, and other growth factors than acute wounds [19]. Plausible explanations are that growth factors might be trapped by the extracellular matrix [20] or that they might be excessively degraded by proteases [21]. Moreover, in chronic wounds, fibroblasts show a decreased potential of proliferation accompanied by an increased number of senescent cells that might impair responsiveness to growth hormones [22].

### 1.2. Malnutrition, Macronutrients, and Chronic Wounds

According to the WHO, malnutrition refers to all forms of deficiency, excess, or imbalance in a person’s intake of energy and/or nutrients [23]. In old people—who are at the highest risk of chronic wounds also due to coexisting diseases—malnutrition often consists of either protein-energy malnutrition or specific vitamin and mineral deficiencies [8]. Several age-related conditions increase the risk of developing nutritional deficiencies, such as clinical, physiological, and socio-economic difficulties that usually affect the elderly [8]. Particularly, in diabetic patients, higher glucose levels could interfere with the process of nutrient absorption, causing the depletion of several nutrients (i.e., magnesium, zinc, B12, B6, folic acid) [24]. While the response to an injury may increase the metabolic needs of the wound area, large amounts of protein can be continually lost through wound exudates [25]. Hence, protein and energy requirements of chronic wound patients may rise by 250% and 50%, respectively [26]. Since cells involved in wound healing require proteins for their formation and activity, protein loss may negatively affect the whole immune process. Proteins are also necessary for immune response, which in turn, if impaired, may delay the progression from the inflammatory to the proliferative phase. In the proliferative and remodeling phases, protein-energy deficiency may also decrease fibroblast activity, delaying angiogenesis and reducing collagen formation [8]. Moreover, protein-calorie deficiency is also associated with weight loss and decreased lean body mass [27]. Hence, implications of weight loss and decreased lean body mass should be recognized when considering the effect of protein-calorie deficiency on the healing process. In general, losing ≈10% lean mass is associated with impaired immunity and increased risk of infection. In case patients lose more than 10% lean body mass, wound healing competes with body demands to restore lean mass: The metabolism gives priority to healing in patients who lose up to 20%, while it delays healing to restore lean body mass in those who lose more than 30% [25,28].

Beyond proteins, both carbohydrates and fats address increased energy needs to support inflammatory response, cellular activity, angiogenesis, and collagen deposition in the proliferative phase of healing process [26]. Particularly, adequate intake of carbohydrates is necessary for fibroblast production and movement, and leukocyte activity [29]. Carbohydrates also stimulate secretion of hormones and growth factors, including insulin that is helpful in the anabolic processes of the proliferative phase. In contrast, hyperglycemia and its complications might reduce granulocyte function and promote wound formation [7]. Fats have structural functions in the lipid bilayer of cell membranes during tissue growth. They are also precursors of prostaglandins—which in turn are mediators of cellular inflammation and metabolism—and participate in several signaling pathways [30]. To date, the effect of supplementation of essential fatty acids on wound healing is controversial. While omega-3 supplementation might decrease wound tensile strength with a harmful effect on healing [31], its combination with omega-6 decreases the progression of pressure ulcers [32]. In line with this evidence, the co-supplementation of omega-3 and omega-6 might lead to benefits, especially during the inflammatory phase [33]. 

### 1.3. Micronutrients and Wound Healing

#### 1.3.1. Amino-Acids

Micronutrients involved in the wound healing process have been extensively reviewed [7,8,9,33]. Among amino-acids, those that play an important role in wound healing, are arginine and glutamine. The first is a precursor of nitric oxide and proline, which in turn are essential for the inflammatory process [34] and synthesis of collagen [35,36]. Arginine also stimulates the production and secretion of growth hormone, as well as the activation of T cells [37,38]. In wound patients with adequate protein intake, the recommended dose of arginine supplementation is 4.5 g/day, while it is useless in the context of protein deficiency [39]. Glutamine plays several roles via its metabolic, enzymatic, antioxidant, and immune properties. In wounds, it protects against the risk of infectious and inflammatory complications by up-regulating the expression of heat shock proteins [40]. Glutamine is also a precursor of glutathione—an antioxidant and an essential cofactor of several enzymatic reactions—which is important for stabilizing cell membranes and for transporting amino acids across them [41]. In addition, glutamine seems involved in the inflammatory phase of wound healing by regulating leukocyte apoptosis, superoxide production, antigen processing, and phagocytosis [40,42]. As for arginine, benefits of glutamine supplementation are still controversial [43] and confounded by the combinations of supplements [44]. 

#### 1.3.2. Vitamins

Vitamins are undoubtedly the most investigated micronutrients in the wound healing process. Vitamin A deficiency impairs B cell and T cell function and antibody production during the inflammatory phase. It also decreases epithelialization, collagen synthesis, and granulation tissue development in the proliferative and remodeling phases [45]. In addition, vitamin A seems to work as a hormone that modulates the activity of epithelial and endothelial cells, melanocytes, and fibroblasts by binding to retinoic acid receptors [46]. In general, vitamin A is topically administered for the care of dermatologic conditions due to its stimulating properties of fibroplasia and epithelialization [33]. In wound patients, it has been recommended to have a short-term supplementation of 10,000–25,000 IU/day to avoid toxicity [33]. Interestingly, vitamin A supplementation counteracts the delay in wound healing caused by corticosteroids for the treatment of inflammatory diseases [47] by down-regulating *TGF-β* and insulin-like growth factor-1 (*IGF-1*) [48]. B vitamins, which consist of thiamine, riboflavin, pyridoxine, folic acid, pantothenate, and cobalamins, are essential cofactors in enzyme reactions involved in leukocyte formation and in anabolic processes of wound healing. Among these, thiamine, riboflavin, pyridoxine and cobalamins are also required for the synthesis of collagen [25]. Hence, vitamin B deficiencies indirectly affect the wound healing process by impairing antibody production and white blood cell function, which in turn increase the risk of infectious complications [49]. Vitamin C seems to be involved in wound healing with several roles in cell migration and transformation, collagen synthesis, antioxidant response, and angiogenesis.

In the inflammatory phase, it participates in the recruitment of cells to the wound and their transformation into macrophages [29]. During collagen synthesis, vitamin C forms extra-bounds between collagen fibers that increase stability and strength of collagen matrix [8]. Vitamin C is essential to counteract the production of free radicals in damaged cells, while its deficiency might increase the fragility of new vessels [50]. The current recommendation of vitamin C supplementation ranges from 500 mg/day in non-complicated wounds to 2 g/day in severe wounds [33]. However, vitamin C supplementation seems to have a beneficial effect only in combination with zinc and arginine, and in pressure ulcer patients [51]. Vitamin D and its receptor (i.e., VDR)—which is ubiquitously expressed in several tissues—modulate structural integrity and transport across epithelial barriers [52]. In line with its roles, recent evidence of vitamin D deficiency among venous and pressure ulcer patients has suggested the potential involvement of vitamin D in the wound healing process [53,54]. However, further research is recommended to understand how vitamin D supplementation might be used in wound care. Although most vitamins show beneficial effects in wound healing, vitamin E might negatively affect collagen synthesis, antioxidant response, and the inflammatory phase [55]. Moreover, vitamin E appears to counteract the benefits of vitamin A supplementation in wound management [56]. 

#### 1.3.3. Minerals

Several minerals are involved in the wound healing process due to their roles as enzyme structural factors, metalloenzymes, and antioxidants. Among these, zinc is essential for DNA replication in cells with high cell division rates, such as inflammatory and epithelial cells, and fibroblasts. In the inflammatory phase, zinc promotes immune response and counteracts susceptibility to infectious complications by activating lymphocytes and producing antibodies [30]. In the proliferative and remodeling phases, it is essential for collagen production, fibroblast proliferation, and epithelialization by stimulating the activity of involved enzymes [8]. Although zinc supplementation of 40–220 mg/day for 10–14 days [57] might be useful in zinc-deficient patients, its benefits in non-deficient patients are currently under debate [9]. Interestingly, topical administration of zinc to surgical wounds significantly improves the healing process [58]. In contrast, conditions that affect zinc metabolism and potential drug-nutrient interactions should be considered for the management of wound patients with zinc supplementation [58]. Less evidence exists on the beneficial effects of iron supplementation for promoting wound healing. As iron transports oxygen to the tissues, it is essential for tissue perfusion and collagen synthesis. Hence, iron deficiency results in tissue ischemia, impaired collagen production, and decreased wound strength in the proliferative phase [30]. 

### 1.4. Curcumin and Wound Healing

In 1910, Milobedzka and colleagues described for the first time the structure of curcumin (Figure 3), one of the three curcuminoids extracted from the powdered rhizome of turmeric plant (*Curcuma longa*) [59]. More recently, it has been demonstrated that curcumin might modulate physiological and molecular events involved in the inflammatory and proliferative phases of the wound healing process [60].

#### 1.4.1. Effects on the Inflammatory Phase

With respect to the inflammatory phase, several studies have revealed the protective effect of curcumin that reduces the expression of pro-inflammatory cytokines, such as tumor necrosis factor alpha (*TNF-α*) and interleukin-1 (*IL-1*) [61]. Accordingly, curcumin recruits M2-like macrophages into white adipose tissues, thereby increasing the production of anti-inflammatory cytokines that are essential for the inflammatory response [62]. In addition, curcumin also inhibits nuclear factor κB (NF-κB) by suppressing the activity of kinases (i.e., AKT, PI3K, IKK) involved in several pathways. In general, NF-κB is physiologically inactivated by binding to its inhibitor IκB. During inflammation, the up-regulation of inflammatory mediators (i.e., cytokines and chemokines) activates NF-κB, which in turn translocates to the nucleus [63]. In wounded sites, curcumin might reduce inflammation caused by the activation of the NF-κB pathway [64]. The anti-inflammatory effects of curcumin are also involved in other signaling pathways, such as peroxisome proliferator-activated receptor-gamma (PPAR-γ) and myeloid differentiation protein 2-TLR 4 co-receptor (TLR4-MD2) [65,66,67,68]. Li and colleagues have reported that curcumin suppresses proliferation of vascular smooth muscle cells by increasing PPAR-γ activity to mitigate angiotensin II-induced inflammatory responses [67]. Additionally, it has been shown that curcumin reduces inflammation through competition with LPS for binding on MD2, thereby inhibiting the TLR4-MD2 signaling complex [68].

Since NF-κB has also several anti-oxidant targets, in 2004, Frey and Malik proposed a relationship between inflammation and oxidation during the wound healing process [69]. In wounds, ROS formation triggers the production and activity of various immune cells (i.e., T lymphocyte subsets, macrophages, dendritic cells, B lymphocytes, and natural killer cells). Moreover, prolonged high ROS concentrations are dangerous for cell structures leading to oxidative stress [70,71]. Particularly, hydrogen peroxide (H_2_O_2_) and superoxide (O_2_^−^) can be considered as potential markers for the amount of oxidative stress [72]. Although anti-oxidant enzymes (i.e., superoxide dismutase, glutathione peroxidase, and catalase) protect cells against toxic ROS levels [73], the imbalance between ROS concentrations and antioxidant activity could determine chronic diseases. Beyond its anti-inflammatory properties, curcumin also acts as an antioxidant by scavenging ROS, by restoring abnormal changes induced by external factors, and by suppressing transcription factors related to oxidation [74,75]. In vitro and in vivo studies have demonstrated the antioxidant activities of curcumin conferred by its electron-donating groups (i.e., the phenolic hydroxyl group) [76]. Moreover, it contributes to the production and activity of antioxidant enzymes [77,78] and their constituents, such as glutathione (GSH) [79]. In line with these findings, Phan and colleagues have revealed the protective role of curcumin against hydrogen peroxide in keratinocytes and fibroblasts [80].

#### 1.4.2. Effects on the Proliferative and Remodeling Phases

As discussed below, curcumin also plays a critical role during the proliferative phase. Interestingly, Gopinath and colleagues have observed that curcumin ameliorates the above-mentioned process, resulting in an increase of hydroxyproline and collagen synthesis [74]. This is consistent with previous studies demonstrating that curcumin decreases the amount of membrane matrix metallo-proteinases (MMPs), which are usually higher in endometriotic mice and human ovarian endometriotic stromal cells. These pathological conditions, in fact, share some molecular events with wound healing, including adhesion and proliferation, cellular invasion, and angiogenesis. Particularly, curcumin could be involved in the process of endometriosis by decreasing the growth and number of endometriotic stromal cells [81]. With respect to wounds, Panchatcharam and colleagues have demonstrated that collagen fibers could mature earlier when wound rats are topically treated with curcumin [70]. Although curcumin does not seem to be involved in the migration of fibroblasts to the wound area in vitro [17], an in vivo study has suggested that curcumin mediates the infiltration of fibroblasts into wound sites, which in turn naturally differentiates into myofibroblasts during the formation of granulation tissue [82]. This controversy might be due to difficulties in creating an in vitro model of fibroblast migration in wounds. Treatment with curcumin also promotes the differentiation of fibroblasts into myofibroblasts [83,84,85,86] which marks the beginning of wound contraction [87]. A previous study has also demonstrated that curcumin reduces the epithelialization period of treated wounds if compared with the control group [70]. Finally, once closure of the wound has been achieved, apoptotic processes discard inflammatory cells from wound sites [88,89,90]. Since curcumin induces apoptosis during the early phase of wound healing, it could also accelerate the healing process by shortening the inflammatory phase [85].

## 2. Discussion

Our review summarizes current evidence about the main biochemical and molecular effects of nutrition, in terms of quality and quantity, on the wound healing process. In line with the Special Issue “Curcumin in Health and Disease”, we have focused on the beneficial effects of curcumin, which exerts its anti-inflammatory and antioxidant properties during the different phases of the wound healing process [11]. Several lines of evidence from in vitro and in vivo studies have reported that curcumin might modulate physiological and molecular events during the inflammatory phase [60,61,65,66,67,68,85]. Moreover, it also exerts antioxidant effects by restoring the imbalance between ROS production and antioxidant activity [74,75,76,77,78,79,80]. In the proliferative phase, curcumin might facilitate collagen synthesis [70,74], fibroblasts migration [82], and differentiation [83,84,85,86]. In addition, curcumin appears to be beneficial for epithelialization [70] and for apoptotic processes that discard inflammatory cells from the wound site [88,89,90]. An in vivo study has suggested that curcumin mediates the infiltration of fibroblasts into wound sites, which in turn naturally differentiates into myofibroblasts during the formation of granulation tissue [82]. By contrast, curcumin does not seem to be involved in the migration of fibroblasts to the wound area in vitro [17].

This controversy might be due to difficulties in creating an in vitro model of fibroblast migration in wounds. In fact, fibroblast migration depends on several factors that cannot be entirely mimicked with in vitro models, such as cell-environment interactions and homeostatic mechanisms [17]. Recently, in wounds of diabetic rats, it has been demonstrated that topical curcumin treatment enhances angiogenesis, thereby ameliorating the healing process [91]. In line with these findings, curcumin could be considered an interesting phytochemical candidate for the treatment of non-healing wounds. Interestingly, its pleiotropic effect on several signaling pathways—by modulating cellular regulatory systems, such as NF-κB, AKT, growth factors, and Nrf2 transcription factor [92,93,94,95]—might be explained by its well-established role in epigenetic mechanisms, such as DNA methylation and histone modification [96]. An understanding of epigenetic regulation in the wound healing process is now becoming an attractive field of research [97], and more efforts should be made to uncover mechanisms underpinning beneficial effects of curcumin and other polyphenols [96,98]. As mentioned above, however, most of these findings come from in vitro or in vivo investigations, while evidence from epidemiological studies is scarce. Given its hydrophobicity and extensive first-pass metabolism [99,100], topical administration of curcumin has a greater effect on wound healing than oral administration [64,88,89]. Despite strides which have been made in the formulation of curcumin for topical application at the wound site [74,83,84,85,101], further research is recommended to improve curcumin delivery and to evaluate its effects in wound patients.

Beyond assessing the potential of curcumin as a wound healing agent, we have also indicated that nutritional assessment in patients at risk of chronic wounds could be the first step towards the prevention of non-healing wounds. In fact, these patients often exhibit protein-energy malnutrition or specific vitamin and mineral deficiencies [8]. The wound healing process, for its part, increases the needs of calories and proteins of the wound area, thereby increasing the requirements from chronic wound patients [26]. Given that protein-calorie deficiencies are further associated with weight loss and decreased lean body mass [27], their implications for wound patients should be also recognized. To meet the increased need of energy, especially during the proliferative phase, wounds also metabolize carbohydrates and fats [26], which in turn are necessary for fibroblast and leukocyte activities, secretion of hormones and growth factors, and structural functions [29,30]. Despite this evidence, the effect of macronutrient supplementation is currently controversial, raising the need for further research. For instance, it has been demonstrated that omega-6 supplementation decreases the progression of pressure ulcers [32], and its combination with omega-3 has beneficial effects on the inflammatory phase [33]. However, omega-3 supplementation alone has harmful effects on healing [31].

Beyond macronutrients, several micronutrients play a crucial role in the wound healing process, as extensively reviewed [7,8,9,33]. Arginine and glutamine exhibit several metabolic, enzymatic, antioxidant, and anti-inflammatory properties that are involved in the inflammatory phase [34,37,38,40,42] and in collagen synthesis [35,36]. However, the beneficial effect of the supplementation of glutamine and arginine, alone or in combination, is still controversial [43,44], probably due to differences in study design, patient characteristics, and type of supplementation. Most of the evidence comes from research on vitamins, with several lines of evidence supporting the benefits of vitamin A [33,47,48], vitamin B [49], vitamin C [8,29,50] and vitamin D [53,54] supplementation. However, to what extent they support wound healing process remains unclear until now. For instance, vitamin C seems to act only in combination with zinc and arginine [51], while vitamin E appears to counteract the benefits of vitamin A [56]. Among minerals, zinc is essential for the inflammatory, proliferative, and remodeling phases by promoting immune response, collagen production, fibroblast proliferation, and epithelialization [8]. Accordingly, topical zinc administration to surgical wounds significantly facilitates wound healing process [58]. These findings cumulatively suggest that nutritional approaches might be useful in the treatment of wounds, especially of chronic non-healing wounds [10]. However, benefits in non-deficient patients are currently under debate [9], and further research should take into account conditions that affect nutrient metabolism, such as diabetes and potential nutrient–nutrient interactions [58].

## 3. Conclusions

In conclusion, we support the notion that curcumin could be considered as a wound healing agent, especially if topically administered. However, most of the current knowledge is derived from in vitro and in vivo investigations, while studies in wound patients remain scarce or controversial. Moreover, since nutrition and nutrients in general might affect the wound healing process, nutritional assessment of patients at risk of non-healing wounds could be the first step towards prevention and treatment. However, further research is recommended to develop appropriate nutritional approaches for wound management.

## Figures and Tables

**Figure 1 ijms-20-01119-f001:**
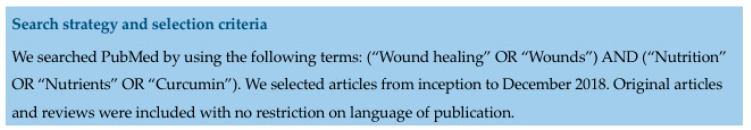
Search strategy and selection criteria.

**Figure 2 ijms-20-01119-f002:**
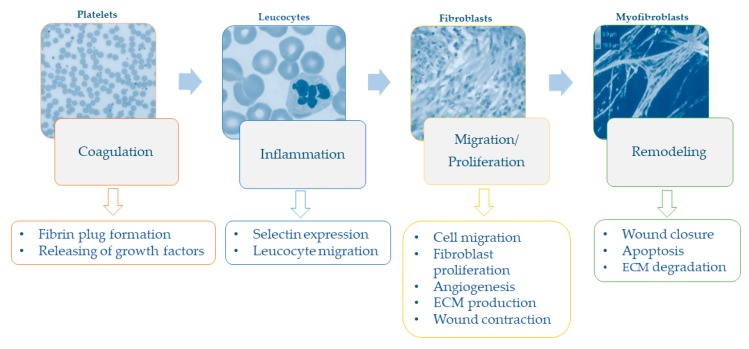
Phases and specific events of the wound healing process.

**Figure 3 ijms-20-01119-f003:**
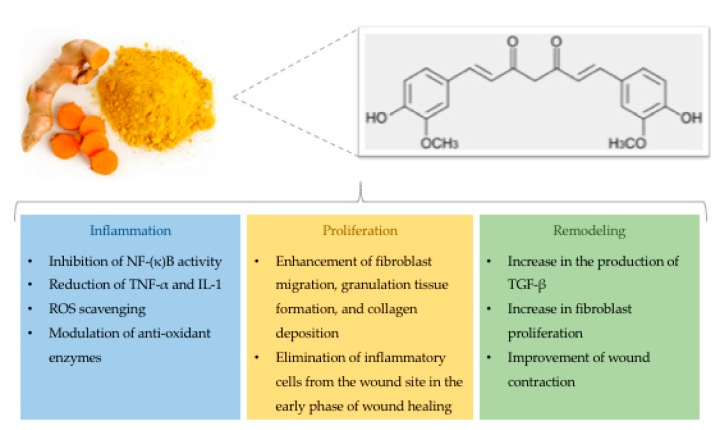
Structure and effects of curcumin on wound healing.

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
