# Peer review of "Nutrition and Wound Healing: An Overview Focusing on the Beneficial Effects of Curcumin"

_ijms, 2019, doi:10.3390/ijms20051119_

Reviewer 1 Report

This is an interesting review of nutrition and wound healing; however, if it is meant to focus on curcumin it is lacking. Curcumin is barely described until the very last section. Multiple other nutrients are described first, and there is no tie to curcumin. If you mean to focus on curcumin and wound healing, this work also misses literature in the field of the endometrium/endometriosis. While this is not described as wound healing, the endometrium regenerates every month. Curcumin has been used and effects on several MMPs have been described. MMPs play a large role in wound healing.

In the intro, I am confused by the numbers given. Statistics are given for the United States; however, the costs are not give in USD. Is this due to the source of the information, the journal, or are these unrelated? US patient info and monetary info on a global or other scale?

Lastly, while overall well written, the English needs to be improved. Mismatch of tenses and use of the word 'the' are noticeable; correction would greatly improve the paper. There are several instances though where the incorrect words are used. For example, 'curcumin is able to determine apoptosis' is incorrect. As this is in the abstract, it sets up the reader to notice all of the other issues.

Author Response

Dear Editor,

thank you very much for considering our manuscript, and for comments of independent Reviewers. We submit to your attention a revised version of the manuscript, in which we have considered all comments. The following List of change and answers to Reviewers addresses all changes made in the manuscript using the track changes function in Microsoft Word.

Reviewer 1

This is an interesting review of nutrition and wound healing; however, if it is meant to focus on curcumin it is lacking. Curcumin is barely described until the very last section. Multiple other nutrients are described first, and there is no tie to curcumin. If you mean to focus on curcumin and wound healing, this work also misses literature in the field of the endometrium/endometriosis. While this is not described as wound healing, the endometrium regenerates every month. Curcumin has been used and effects on several MMPs have been described. MMPs play a large role in wound healing.

We are grateful to Reviewer 1 for his/her comments. As mentioned in the introduction section, our aim was to provide an overview on the relationship between nutrition and wound healing, with a particular focus on the role of curcumin. Thus, we first described molecular events of wound healing process and the effects of main nutrients. Next, we focused on the beneficial properties of curcumin. According to Reviewer suggestions, we placed more emphasis on the effects of curcumin in the introduction and discussion sections. Moreover, we included some relevant references on the potential role of curcumin in endometriosis (please see lines 67-72 and 267-270).

In the intro, I am confused by the numbers given. Statistics are given for the United States; however, the costs are not give in USD. Is this due to the source of the information, the journal, or are these unrelated? US patient info and monetary info on a global or other scale?

We are sorry for the mistakes. In this new revised version of our manuscript we provide, as an example, consistent statistics for United States.

Lastly, while overall well written, the English needs to be improved. Mismatch of tenses and use of the word 'the' are noticeable; correction would greatly improve the paper. There are several instances though where the incorrect words are used. For example, 'curcumin is able to determine apoptosis' is incorrect. As this is in the abstract, it sets up the reader to notice all of the other issues.

We are grateful for positive comments by Reviewer 1 and for his/her helpful suggestions. According to his/her comments, we corrected English style and language to improve quality of our manuscript.  

Reviewer 2 Report

the ms by barchitta et al represent a good work in the field, but before acceptance there are some points to clarify. in particular, i suggest to highlitght the effects of curcumin on single skin cells, such as keratinocytes and fibroblasts, as well as to modfiy discussione and conclusion section in order to put more emphasis on the role of curcumin for wound healing and tissue regeneration.

Author Response

Dear Editor,

thank you very much for considering our manuscript, and for comments of independent Reviewers. We submit to your attention a revised version of the manuscript, in which we have considered all comments. The following List of change and answers to Reviewers addresses all changes made in the manuscript using the track changes function in Microsoft Word.

Reviewer 2

the ms by barchitta et al represent a good work in the field, but before acceptance there are some points to clarify. in particular, i suggest to highlitght the effects of curcumin on single skin cells, such as keratinocytes and fibroblasts, as well as to modfiy discussione and conclusion section in order to put more emphasis on the role of curcumin for wound healing and tissue regeneration.

We are grateful to Reviewer 2 for his/her positive comments. As suggested, in the discussion and conclusion sections, we placed more emphasis on the role of curcumin in wound healing process, also including relevant references about its effects on fibroblasts (please see lines 296-306).  

Reviewer 3 Report

This is a comprehensive and concise review regarding malnutrition, macro/micro-nutrients and wound healing and the beneficial effects of curcumin on wound healing.  There are some minor issues needed to be addressed:

1: Angiogenesis is a critical part of wound healing as described in the text. However, it is not demonstrated in Fig 2. 

2: Discuss more about how diabetes affects the metabolism and absorption of nutrients.

3: In section of Curcumin and wound healing, how curcumin affects wound healing at different stages should be discussed under different subtitles like Fig 3, which will make it easier for readers to follow. 

Author Response

Dear Editor,

thank you very much for considering our manuscript, and for comments of independent Reviewers. We submit to your attention a revised version of the manuscript, in which we have considered all comments. The following List of change and answers to Reviewers addresses all changes made in the manuscript using the track changes function in Microsoft Word.

Reviewer 3

This is a comprehensive and concise review regarding malnutrition, macro/micro-nutrients and wound healing and the beneficial effects of curcumin on wound healing.  There are some minor issues needed to be addressed:

1: Angiogenesis is a critical part of wound healing as described in the text. However, it is not demonstrated in Fig 2. 

We are grateful to Reviewer 3 for his/her positive comments. As requested, we included angiogenesis in Figure 2.

2: Discuss more about how diabetes affects the metabolism and absorption of nutrients.

According to Reviewer 3 suggestion, we included a relevant reference about the effect of diabetes on nutrient absorption (please see lines 123-125).

3: In section of Curcumin and wound healing, how curcumin affects wound healing at different stages should be discussed under different subtitles like Fig 3, which will make it easier for readers to follow. 

As suggested, we used different subtitles in the 1.4 section.

Round  2

Reviewer 1 Report

The information on curcumin is much improved. Before it felt as almost an afterthought, now it is well laid out and more informative.